# Indicators of Eating Disorders in Sexually Abused Brazilian Adolescents: Family and School Contexts

**DOI:** 10.3390/children10081393

**Published:** 2023-08-15

**Authors:** Julia Altoé Andrade, Luciane Bresciani Salaroli, Priscilla Rayanne e Silva Noll, Matias Noll, Sheila Oliveira Feitosa, Rodrigo Daminello Raimundo, Adriana Gonçalves de Oliveira, Carolina Rodrigues Mendonça, Luiz Carlos de Abreu

**Affiliations:** 1Nutrition and Health Postgraduate Program, Health Sciences Center, Federal University Espirito Santo, Vitória 29075-910, Brazil; julialtoe@hotmail.com (J.A.A.); lucianebresciani@gmail.com (L.B.S.); 2Instituto Federal Goiano, Ceres 76300-000, Brazil; priscilla.silva@ifgoiano.edu.br (P.R.e.S.N.); matias.noll@ifgoiano.edu.br (M.N.); sheilaoliveirafeitosa@gmail.com (S.O.F.); 3Health Sciences Postgraduate Program, Medical School, Universidade Federal de Goiás, Goiânia 74605-080, Brazil; carol_mendonca85@hotmail.com; 4Laboratório de Delineamento de Estudos e Escrita Científica, Centro Universitário FMABC, Santo André 09060-590, Brazil; rodrigo.raimundo@fmabc.br (R.D.R.); agdeoliveira@gmail.com (A.G.d.O.)

**Keywords:** eating disorders, laxatives, adolescents health, sexual violence, sexual crimes, sexual assault victims, family context, school context, bullying

## Abstract

Eating disorders, characterized by abnormal eating behaviors, are among a wide variety of psychiatric conditions that mainly affect children and adolescents. These disorders have a multifactorial origin and can be associated with restrictive diets, negative feelings, harmful family relationships, and post-traumatic stress. Thus, this study’s objective was to evaluate the association between indicators of eating disorders and family and school contexts in Brazilian adolescents who previously experienced sexual abuse and examine the findings based on sex. National School Health Survey data were utilized. Among 102,301 students between 11 and 19 years of age, 4124 reported having experienced sexual abuse and were included in this study. Self-report questionnaires were used to assess participants’ health status and the presence of risk behaviors, which were examined through multivariate analysis using a Poisson regression model. The results indicated positive relationships between self-induced vomiting, laxative misuse, and other purgative methods and infrequent meals with family, hunger, and the presence of violence in students’ daily lives, regardless of sex (*p* < 0.05). In addition, body dissatisfaction and negative feelings about one’s body were associated with having been bullied or teased by schoolmates for both sexes (*p* < 0.05). Distant relationships with parents were associated with purgative methods and body dissatisfaction among female students (*p* < 0.05). In conclusion, body dissatisfaction, negative feelings about one’s body, laxative misuse, self-induced vomiting, and purgative methods were found to be associated with factors in family and school contexts such as hunger, infrequent meals with family, family violence, distant relationships with parents, and bullying at school in adolescents who have previously experienced sexual abuse.

## 1. Introduction

Eating disorders (EDs) are serious psychiatric illnesses that primarily affect children and adolescents [1,2,3], and are characterized by various abnormal eating behaviors such as restricted food intake, use of drugs and other substances harmful to health, voluntary purgative [4,5] methods, and binge eating [4,6]. The fifth edition of the *Diagnostic and Statistical Manual of Mental Disorders*, classifies anorexia nervosa (AN), bulimia nervosa (BN) and binge eating disorder as the main EDs [5]. Although the exact worldwide prevalence of EDs is unknown, a recent systematic review analyzing 121 studies published between 2000 and 2018 on the epidemiology of EDs in the general population reported the weighted average prevalence in the global population in terms of data from broad categories of EDs was 17% [7]. In general, research has indicated that the prevalence of EDs is higher in women and young people than in men or other age groups [8,9].

Self-induced vomiting, laxative misuse, and other purgative methods typically begin or intensify during adolescence, at approximately 15–25 years of age, and are associated with decreased quality of life [2,10]. Body dissatisfaction; overeating; restrictive dieting; negative feelings such as guilt, sadness, fear, and anxiety; and parental influence may indicate an ED, especially if two or more are present [11,12,13]. Negative life events such as bullying, parental divorce, oppressive relationships with parents, bereavement, and early trauma such as sexual abuse may be associated with or predisposing factors for ED development [14,15,16]. These events are usually traumatic, and affected individuals may choose to decrease or increase eating behavior in an attempt to assuage, control, or exclude traumatic memories [15,17].

Eating habits can become a method of blocking unwanted emotions or expressing hatred toward one’s own body [18]. Such childhood events leave negative memories that are associated with several comorbid psychiatric disorders, including EDs [19]. Thus, some recent literature has focused on evaluating the association between child abuse and symptoms indicative of EDs, especially physical and sexual abuse [15,20]. However, studies on this topic remain scarce.

Aspects of the environment in which children and adolescents develop can influence how they approach life and its problems. School is where many of adolescents’ social interactions occur and can significantly contribute to the mental health of this population [21]. Mental health risk factors for adolescents in the school setting consist of bullying, academic failure, and the inability to provide adequate settings for student learning [22]. Violence can also occur in schools and may be related to physical and mental health disorders in adolescents, which can be considered a public health problem [15,23].

EDs and sexual abuse may have an intense and permanent impact on adolescent physical and mental health [24,25]. To date, few studies have aimed to verify associated factors or indicators of EDs in adolescents in family and school contexts using a representative national sample [24]. Several studies have been limited to small clinical or community samples, included only one type of EDs, or been aimed toward only the female population, making it difficult to obtain information on this topic [15]. Understanding the factors associated with indicators of EDs related to family and school contexts is essential to better understand sexual abuse survivors and provide them with effective treatment for resulting in positive long-term mental health outcomes [26]. Nevertheless, supporting health-promoting actions and improving adolescents’ quality of life according to sex differences remains important. Therefore, our study aimed to evaluate the association between ED indicators and family and school contexts in Brazilian adolescents who have experienced sexual abuse and examine these associations based on sex.

## 2. Methods

### 2.1. Participants and Study Period

This cross-sectional study used data from the third edition of the National School Health Survey (PeNSE). The third edition of the PeNSE was conducted in 2015 by trained technicians from the Brazilian Institute of Geography and Statistics and evaluated students who were attending their ninth year of school in Brazil [27,28]. This study was approved by the National Research Ethics Committee (protocol no. 1,006,467). The sample size was calculated based on all 5 Brazilian regions (South, North, Midwest, Southeast, and Northeast), considering the 26 states and Federal Districts as well as their municipalities, on an equiprobability basis [27].

### 2.2. Sample Selection and Data Collection

The sample included 102,301 students regularly enrolled across 3040 schools, divided into 4159 classes, who were in class at the moment of data collection. However, the present study only included students who reported previous sexual abuse (*n* = 4124, 4.1%). Data were obtained by a self-report questionnaire administered to public and private students from schools throughout Brazil [27]. All scholars from the selected classes were asked to complete the questionnaire using smartphones. Students were also advised that they could refuse to participate at any time during the study and were guaranteed anonymity and privacy [27,28].

The questionnaire includes questions about socioeconomic status, diet, physical activity levels, and risk behaviors (e.g., tobacco use, consumption of alcoholic beverages and illicit drugs, situations at home and school, mental health, sexual and reproductive health, safety, body image, and use of health services). In this sense, several studies have been dedicated to evaluating health aspects and risk behaviors [21,29,30,31,32,33,34].

The outcomes evaluated were indicators of behavioral clusters of EDs assessed based on responses to the following questions: (1) in the last 30 days, have you vomited or taken laxatives to lose weight or avoid gaining weight (Yes or No); (2) in the last 30 days, have you taken any medications, formulas, or other weight loss products without medical supervision (Yes or No); (3) how do you feel about your body (very satisfied, satisfied, indifferent, dissatisfied, or very dissatisfied); and (4) as for your body, do you consider yourself very thin, thin, normal, fat, or very fat? The answers to Questions 3 and 4 were grouped into dichotomous variables, resulting in “very satisfied, satisfied, or indifferent” or “dissatisfied or very dissatisfied” for Question 3 and “very thin, thin, or normal” or “fat or very fat” for Question 4.

The following questions related to family and school contexts were adopted as independent variables: “In the last 30 days, how often were you hungry because you did not have enough food in your home”; “Do you usually have lunch or dinner with your mother, father, or guardian”; “In the last 30 days, how often did your parents or guardians understand your problems and concerns”; “In the last 30 days, how many times were you physically assaulted by an adult in your family”; “In the past 30 days, how often did any of your schoolmates bully or tease you so much that you were hurt, annoyed, offended, or humiliated”; and “In the last 30 days, what was the reason/cause that your schoolmates teased, mocked, bullied, or humiliated you”? Sociodemographic characteristics (sex, age group, and mother’s educational background), Brazilian region, and school type (public/private) were considered also as adjustment variables.

### 2.3. Data Analysis

Dependent variables used in the data analysis included self-induced vomiting and/or laxative misuse, use of medication or weight loss products, body dissatisfaction, and negative feelings about one’s body, which characterize the behavioral clusters common in EDs. Socioeconomic and school- and family-related variables were considered independent variables. Dependent variables were evaluated by descriptive and inferential statistics (Wald’s chi-square association). Independent variables were examined by a Poisson regression model set up by robust variance (crude analysis).

Prevalence ratios (PRs) with their respective 95% confidence intervals (CIs) were used as the measure of effect. Crude PRs were obtained using unadjusted Poisson regression [35]. The analysis was adjusted for sociodemographic variables. These adjustment variables were selected based on potential confounders as supported by previous methodological and statistical studies [35,36,37]. SPSS 26.0 (IBM Corp., Armonk, NY, USA) was used for all data analysis.

## 3. Results

Of the 102,301 students who participated in the PeNSE, 101,300 answered questions regarding previous abuse. Adolescents who reported having previously experienced sexual abuse (*n* = 4124; 4.10%) were included in the present study, of whom 42% (*n* = 1734) were male and 58% (*n* = 2390) were female. Practical variables of vomiting and/or consumption of laxatives, use of medication or weight loss products, body dissatisfaction, and negative feelings about one’s body were associated with male sex and age. These variables were also associated with all included socioeconomic variables, except negative feelings about one’s body and ethnicity, and the practice of vomiting and/or consumption of laxatives, region, school type, and use of weight-loss medication (Table 1).

The practice of vomiting and/or consumption of laxatives was associated with students of both sexes who were often hungry because they did not have sufficient food at home and had been physically assaulted by an adult family member in the last month (Table 2). Having lunch or dinner with one’s parents or guardians up to four times per week was associated with more self-induced vomiting and laxative consumption in female students. Male students who were bullied or humiliated by their schoolmates because of their physical appearance reported less self-induced vomiting and laxative consumption.

The use of medication or weight loss products was associated with students of both sexes who were often hungry because they did not have sufficient food at home, female students who had meals with their parents or guardians up to four times per week, and students of both sexes who had been physically assaulted in the previous month by an adult family member. Being understood by those guardians in relation to their problems was associated with the use of medication or weight loss products in female students; however, the opposite was found among male students. In the school context, female students who were bullied or teased by classmates in the previous month reported more use of medication or weight loss products (Table 3).

Body dissatisfaction was associated with meals with one’s parents/guardians less than once per week, regardless of sex. For female students, not feeling understood by one’s parents and being physically assaulted four or more times in the last month by an adult family member were also associated with greater body dissatisfaction (Table 4). Being bullied or teased by schoolmates was the factor most often associated with body dissatisfaction regardless of sex. Body or face appearance and sexual orientation were also factors associated with body dissatisfaction.

For students of both sexes, hunger due to insufficient food at home, infrequent meals with their parents or guardians, being physically assaulted, or being bullied or teased by their schoolmates were associated with negative feelings about one’s own body (Table 5). Moreover, being teased by schoolmates because of one’s physical appearance was associated with negative feelings about one’s body. Female students who rarely felt understood by their parents or guardians regarding their problems also reported more negative feelings about their bodies.

After the adjusted analysis, the use of laxatives and self-induced vomiting remained associated with frequent hunger and having been assaulted by an adult family member in the last month in both male and female students (Table 6). Both sexes were associated with being teased by schoolmates. Having meals with one’s parents or guardians and being understood by them in relation to one’s problems were associated with reports of self-induced vomiting and/or use of laxatives in female students, whereas the reverse was found for male students.

The misuse of medication or weight loss products remained associated with family and school context variables, with the exception of female students who were often hungry and male students who were not understood by their parents regarding their problems.

For both sexes, body dissatisfaction remained associated with rarely having lunch or dinner with one’s parents and being bullied or teased by schoolmates due to physical appearance (Table 7). Female sex remained associated with body dissatisfaction, not being understood by parents or guardians, and aggression from family members in the last 30 days. After adjusting for confounders, negative feelings about one’s body were found to be associated with frequent feelings of hunger in the last month, having meals with guardians one day at most in the last week, and having been bullied by peers in the last month because of one’s physical appearance. In female students, these negative feelings were associated with not being understood by one’s parents or guardians and having been assaulted by an adult family member in the last month.

## 4. Discussion

The surveys conducted as part of the PeNSE aimed to better understand adolescents’ way of life and behavior through questions about their daily lives and eating habits in addition to the socioeconomic and cultural context in which the students lived. The analyses were intended to contribute to the planning of public policies and health actions. Using the PeNSE database, this study found that, in both male and female students, self-induced vomiting, laxative misuse, and other purgative methods were associated with infrequent meals with one’s parents or guardians, frequent hunger, and the presence of violence in students’ daily lives. Body dissatisfaction and negative feelings about one’s body were also associated with bullying by schoolmates, the presence of violence, infrequent meals with one’s parents or guardians, and frequent hunger in both sexes. For female students, a distant relationship with one’s parents was associated with purgative methods and body dissatisfaction.

This study demonstrated that those who experience humiliation from their schoolmates or abuse from a family member often present with some type of ED indicator. This study also addressed the role of sexual violence in the emergence of EDs in adolescents. In addition, this study used a large database with information collected nationwide in both public and private schools, and with the entire age group that comprises the period of adolescence.

Many people suffer several traumatic experiences over the course of their lives, especially during adolescence, and many of them experience trauma in environments that should provide them with comfort and care, such as in their own homes and in school. These traumas can make adolescents psychologically vulnerable and, consequently, at risk of developing an ED [15]. The population in this study had reported previously experiencing sexual abuse, which is an intense trauma and intermediate variable for EDs [24,38,39,40,41]. A recent meta-analysis that evaluated 29,272 individuals with EDs and 1,679,385 controls noted evidence suggesting prior sexual abuse as a risk factor for BN and an association between appearance-related teasing victimization and any ED [24]. Despite the lack of knowledge regarding the relationship between these factors [24], it is suggested that binge and dissociation behaviors, signs of EDs, are important to trauma survivors [40]. This association has implications for ED prevention and treatment [42].

The most commonly observed EDs in adolescents are AN and BN [43]. Among the adolescents in our research, the highest prevalence of EDs was in female students (59.5%), of whom 31.4% and 28.2% showed signs of AN and BN, respectively. In male students, the prevalence of EDs was 27.4% (11.3% for AN and 16.2% for BN). This study does not relate the presence of EDs to the occurrence of abuse in adolescents; however, it does show that even in those who do not experience these practices, EDs are often present, demonstrating the tendency of this population to develop some type of disorder related to eating or body image [43].

Bullying among children is a substantial public health problem worldwide and involves different levels of aggression, ranging from verbal to physical abuse. Bullying can be direct, which occurs physically or verbally; indirect, through social exclusion and psychological abuse; or cyberbullying, which is an offense committed through digital means. Regardless of when or how bullying occurs, abusers become more aggressive as victims succumb to the vulnerability of problems [44].

In a study conducted with 208 schoolchildren aged 12–14 years, 48.1% had already engaged in some form of bullying, and 34.6% had experienced some type of humiliation. When comparing these data with data on body dissatisfaction in this population, the rate of ATs was found to be higher among bullying victims (71.79%). Among them, male adolescents were the most affected, showing a direct relationship between BMI and body dissatisfaction [45].

The present study also found a relationship between infrequently eating with parents or guardians and purgative practices, especially in female students (PR: 1.35; 1.15–1.59). One factor contributing to this relationship is the type of relationship that daughters have with their mothers and their low education level, which can be considered a risk factor for ED development [15]. Families who are excessively concerned about the physical appearance of daughters and often address issues related to body image contribute to greater body dissatisfaction among adolescents [46]. A study evaluating parenting style and mental disorders in 6483 US adolescents noted that a high level of care, particularly that from mothers, was associated with inferior odds of eating and behavioral disorders, in contrast to high maternal control, which was associated with advanced odds of eating and behavioral disorders beyond depression and anxiety. Paternal and material control combined was associated with anxiety disorders and substance abuse [47].

In the present study, hunger caused by not having sufficient food at home was associated with self-induced vomiting, laxative use and purgative methods, body dissatisfaction, and negative feelings about one’s body in both sexes. Notably, it is a marker of food and nutritional insecurity in households, characterizing itself as a social determinant of health variables [48]. Food and nutritional insecurity has been associated with disordered eating patterns in previous studies [48,49,50].

Several factors are associated with taking extreme measures (e.g., using laxatives and/or medications and inducing vomiting) for weight loss among adolescents [51], among which a lack of family care/support is one of the most frequently reported variables [40,47]. Nevertheless, adolescents who live with their parents, keep them informed about their lives, and frequently have meals with them are 18%, 52%, and 10% less likely to use laxatives, use medications, or induce vomiting, respectively [51].

The study also found that female students were 17% more likely to exhibit self-induced vomiting, laxative misuse, and other purgative methods than male students. Adolescents who perceive themselves as overweight are 142% more likely to adopt any of the abovementioned extreme attitudes because of their degree of body dissatisfaction when compared to those who perceive themselves to be within an appropriate weight range [51]. Other studies have shown that the incidence of EDs is higher in girls who report high processed food consumption. The rates of overweight/obesity and abdominal and body fat are higher in these adolescents who eat more processed foods, making body dissatisfaction, aesthetic pressure, and bullying related to one’s physical appearance more frequent, leading them to develop certain EDs [3]. The opposite can also be observed when bullying or social exclusion triggers inappropriate behavior in relation to food, causing the individual to take out their feelings and frustrations on food, which can lead to becoming overweight or obese, which are possible triggers for the development of EDs [52].

Psychological abuse can also significantly influence the degree of body satisfaction in this population, especially when it comes to female adolescents (the part of this population most negatively affected by abusive practices). This type of violence results in a two- to four-times increase in the chances of individuals being dissatisfied with their body and thus developing an ED [34]. In addition to psychological abuse, physical abuse is a risk factor for increased body dissatisfaction among adolescents; the more severe the violence, the greater the body dissatisfaction [24]. Girls have been characterized as being the most affected by family violence, and those who experience this violence are less likely to become overweight, suggesting that the experience of physical abuse in this population is related to their BMI [34].

The number of cases of abuse, mistreatment, and neglect experienced by children and adolescents, particularly in family and school environments, has increased considerably in recent years. Thus, multidisciplinary and psychopedagogical teams in schools need to seek out and conduct interventions, in addition to including parents or guardians in these EDs actions [53].

Nutrition education seeks to change individuals’ perceptions of the food they eat, helping them change their perspectives on food. However, nutrition professionals must create a bond with patients so that those patients can trust and believe in the change process. In addition, nutrition professionals need to investigate the patient’s past history and understand their motivations without neglecting the interpretation of biochemical and physical examinations and anthropometric measurements (provided that they can be measured without embarrassing the patient) [54].

The present work illuminates various factors that influence the development of EDs, including abuse experienced during adolescence, particularly sexual abuse, which can influence several aspects related to adolescents’ mental health. These findings were only possible because of the rigorous methodology used in this study, which addressed a novel theme from the perspective of EDs and sexual violence. A comprehensive understanding of the indicators of EDs in family and school contexts among adolescents who have experienced sexual abuse may support clinicians and survivors address indicators of disordered eating and weight-related outcomes as well as recognize action strategies for preventing injuries and ensuring necessary treatments [40]. Epidemiological data on adolescents from developing countries such as Brazil can shed light on this topic, reinforcing the need for complex and interdisciplinary public health policies and programs to guide the direction of public health [40]. For example, the link between sexual abuse and EDs has implications for ED treatment and prevention. Therefore, in treatment, EDs should not be emphasized as an individual pathology as this reinforces the original trauma. Instead, the theme should be approached using methods that shape the ability of adolescents to respond to, make sense of, and resist trauma while living healthy lives [39].

Nevertheless, the limitations of this study should also be highlighted. Due to the cross-sectional design, causal relationships between the variables studied cannot be inferred, and patterns and trends in the studied population cannot be determined. Furthermore, self-report questionnaires may have led to bias for some responses. The study also did not include adolescents’ perceived body image as a variable, which could add to the details of the data obtained. Thus, more studies are needed on sexual abuse prevention and its impact on the development of EDs, including the effects of the pandemic on this relationship.

## 5. Conclusions

A relationship exists between indicators of EDs, such as self-induced vomiting, laxative misuse and additional purgative methods, body dissatisfaction, and negative feelings about one body, and low perceived parental care, hunger, and violence experienced at home in adolescents of both sexes. In the school context, body dissatisfaction and negative feelings toward one’s body were associated with having been bullied or teased by schoolmates. The involvement of parents and guardians with these adolescents is a protective factor against both violence and indicators of EDs, particularly in female adolescents. Therefore, adolescents who experience physical, psychological, and sexual abuse may be more dissatisfied with their bodies compared with their peers, which may contribute to the development of EDs.

## Figures and Tables

**Table 1 children-10-01393-t001:** Sociodemographic characteristics of Brazilian adolescents who reported sexual abuse in terms of being indicative of eating disorders.

Sociodemographic Variables	Vomiting and/or Consumption of Laxatives			Medicine or Weight Loss Product			Body Dissatisfaction			Negative Feeling about the Body		
	Total%	%	*p*	Total%	%	*p*	Total%	%	*p*	Total%	%	*p*
**Sex**			<0.001			<0.001			<0.001			<0.001
Male	41.9	35.7		41.7	36.0		41.7	19.6		41.8	21.4	
Female	58.1	20.9		58.3	14.5		58.3	30.4		58.2	35.1	
**Age**			<0.001			<0.001			<0.001			<0.001
≤13 years	10.4	24.1		10.4	19.1		10.3	34.5		10.4	35.6	
14 years	41.9	23.6		42.0	19.7		42.0	26.9		41.9	30.5	
15 years	26.5	29.0		26.4	25.4		26.5	24.9		26.5	29.2	
≥16 years	21.2	33.0		21.2	30.6		21.2	21		21.2	24.4	
**Race/ethnicity**			0.008			<0.001			0.004			0.160
White	27.9	25.9		28.0	22.3		28.0	29.7		27.9	31.3	
Black	15.3	31.7		15.3	30.1		15.2	22.2		15.3	27.0	
Yellow	5.5	32.9		5.5	30.4		5.5	28.3		5.5	33.8	
Brown	46.2	25.5		46.1	21.2		46.1	24.6		46.2	28.5	
Indigenous	5.1	27.4		5.1	22.6		5.1	25.5		5.1	29.3	
**Region**			0.084			0.043			<0.001			<0.001
South	9.1	22.6		9.2	20.3		9.1	33.2		9.1	36.9	
Southeast	14.3	29.5		14.3	25.5		14.2	31.2		14.3	35.7	
Midwest	14.8	29.5		14.8	27.0		14.9	26.2		14.8	28.0	
North	30.8	25.9		30.8	21.7		30.8	23.6		30.8	28.2	
Northeast	31.0	27.3		30.9	23.5		30.9	23.6		30.9	26.1	
School			0.793			0.160			<0.001			<0.001
Public	87.4	27.2		87.3	23.8		87.3	24.3		87.3	27.2	
Private	12.6	26.6		12.7	21.0		12.7	36.9		12.7	44.2	
**Mother’s schooling**			0.002			<0.001			<0.001			<0.001
No one	11.6	35.5		11.6	33.3		11.6	21.1		11.6	23.1	
Elementary school	36.2	27.2		36.1	22.8		36.2	22.5		36.2	25.8	
High school	28.0	24.5		28.1	22.0		28.0	27.5		28.0	32.1	
University education	24.1	28.0		24.2	25.9		24.2	31.8		24.3	35.0	

**Table 2 children-10-01393-t002:** Association between laxative misuse and/or self-induced vomiting and independent variables in adolescents who reported sexual abuse (*n* = 4084).

Variables	Total%	Vomiting and/or Consumption of Laxatives					
		Male			Female		
		Prevalence (%)	RP Gross (IC 95%)	*p*	Prevalence (%)	RP Gross (IC 95%)	*p*
**Hungry because did not have enough food at home (past 30 days)**				0.004			0.005
Never or rarely	72.8	34.3	1		19.3	1	
Sometimes	16.2	35.2	1.02 (0.85–1.24)		24.6	1.27 (1.04–1.55)	
Most of time or always	11.0	48.9	1.42 (1.18–1.72)		27.5	1.42 (1.11–1.81)	
**Having lunch or dinner with parents/guardian**				0.048			<0.001
≥5 days of week	60.1	37.0	1		17.9	1	
1 to 4 days of week	8.7	39.5	1.07 (0.85–1.34)		25.5	1.43 (1.09–1.88)	
<1 day of week, weekly, or never	31.2	30.8	0.83 (0.71–0.98)		24.2	1.35 (1.15–1.59)	
**Parents or guardians understand problems and concerns (past 30 days)**				0.274			0.125
Most of time or always	53.2	38.2	1		18.0	1	
Sometimes	19.2	33.2	0.87 (0.72–1.04)		20.9	1.16 (0.90–1.50)	
Never or rarely	27.6	35.2	0.92 (0.80–1.06)		22.1	1.23 (1.00–1.50)	
**Physically assaulted by an adult at family** **(past 30 days)**				<0.001			<0.001
Not once	38.9	22.3	1		15.8	1	
1–3 times	23.4	39.4	1.77 (1.47–2.13)		25.0	1.57 (1.30–1.90)	
≥4 times	37.7	55.4	2.48 (2.14–2.87)		36.0	2.27 (1.89–2.73)	
**Bullied or bullied by schoolmates** **(past 30 days)**				0.189			<0.001
Never or rarely	37.0	34.6	1		18.2	1	
Sometimes	41.5	35.1	1.01 (0.88–1.17)		18.7	1.03 (0.85–1.24)	
Most of time or always	21.5	40.5	1.17 (0.98–1.39)		34.8	1.91 (1.57–2.34)	
**Reason/cause of colleagues intimidating or humiliating (past 30 days)**				0.002			<0.001
My color or race	12.2	45.8	1		33.0	1	
My religion	8.7	45.1	0.98 (0.72–1.34)		35.4	1.07 (0.69–1.67)	
The appearance of my face	13.3	40.5	0.88 (0.66–1.18)		23.2	0.70 (0.47–1.05)	
The appearance of my body	16.1	32.3	0.70 (0.51–0.97)		26.6	0.81 (0.56–1.16)	
My sexual orientation	9.7	37.2	0.81 (0.60–1.09)		32.2	0.98 (0.61–1.57)	
My region	3.8	53.8	1.17 (0.83–1.67)		27.8	0.84 (0.38–1.88)	
Others	36.3	29.8	0.65 (0.50–0.83)		17.9	0.54 (0.39–0.76)	

**Table 3 children-10-01393-t003:** Association between use of medication, formula, or weight loss product and independent variables in Brazilian adolescents who reported sexual abuse (*n* = 4074).

Variables	Total%	Medicine or Weight Loss Product					
		Male			Female		
		Prevalence (%)	RP Gross (IC 95%)	*p*	Prevalence (%)	RP Gross (IC 95%)	*p*
**Hungry because did not have enough food at home (past 30 days)**				0.001			0.094
Never or rarely	7.3	33.8	1		13.7	1	
Sometimes	15.4	39.8	1.18 (0.99–1.40)		16.1	1.18 (0.91–1.52)	
Most of time or always	8.2	49.2	1.45 (1.20–1.76)		18.7	1.37 (1.01–1.87)	
**Having lunch or dinner with parents/guardian**				0.134			0.008
≥5 days of week	60.3	37.2	1		12.5	1	
1 to 4 days of week	7.5	37.4	1.00 (0.79–1.28)		18.5	1.48 (1.06–2.08)	
<1 day of week, weekly, or never	32.1	31.8	0.85 (0.73–1.00)		16.5	1.32 (1.08–1.63)	
**Parents or guardians understand problems and concerns (past 30 days)**				0.010			0.044
Most of time or always	27.1	40.8	1		11.4	1	
Sometimes	19.5	31.6	0.77 (0.64–0.93)		14.8	1.30 (0.95–1.80)	
Never or rarely	53.4	34.4	0.84 (0.73–0.97)		15.7	1.38 (1.07–1.79)	
**Physically assaulted by an adult at family** **(past 30 days)**				<0.001			<0.001
Not once	57.8	21.5	1		10.8	1	
1–3 times	20.9	39.9	1.86 (1.54–2.24)		16.2	1.50 (1.17–1.90)	
≥4 times	21.3	57.7	2.69 (2.32–3.12)		27.3	2.52 (2.01–3.16)	
**Bullied or bullied by schoolmates** **(past 30 days)**				0.814			<0.001
Never or rarely	39.3	36.8	1		12.4	1	
Sometimes	45.0	35.2	0.96 (0.83–1.10)		12.6	1.02 (0.81–1.28)	
Most of time or always	15.7	36.6	1.00 (0.83–1.19)		25.6	2.06 (1.61–2.65)	
**Reason/cause of colleagues intimidating or humiliating (past 30 days)**				0.002			<0.001
My color or race	8.5	40.8	1		27.3	1	
My religion	6.0	43.9	1.07 (0.78–1.49)		30.8	1.13 (0.68–1.86)	
The appearance of my face	12.4	37.3	0.91 (0.67–1.25)		18.1	0.66 (0.42–1.05)	
The appearance of my body	15.8	44.8	1.10 (0.82–1.47)		20.2	0.74 (0.49–1.12)	
My sexual orientation	7.6	32.3	0.79 (0.57–1.10)		18.6	0.68 (0.36–1.29)	
My region	2.3	50.0	1.22 (0.83–1.80)		16.7	0.61 (0.21–1.81)	
Others	47.5	28.1	0.69 (0.52–0.90)		11.0	0.40 (0.27–0.60)	

**Table 4 children-10-01393-t004:** Association between body dissatisfaction and independent variables in Brazilian adolescents who reported sexual abuse (*n* = 4078).

Variables	Total%	Body Dissatisfaction					
		Male			Female		
		Prevalence (%)	RP Gross (IC 95%)	*p*	Prevalence (%)	RP Gross (IC 95%)	*p*
**Hungry because did not have enough** **food at home (past 30 days)**				0.081			0.672
Never or rarely	76.4	19.5	1		30.5	1	
Sometimes	15.3	16.7	0.86 (0.63–1.17)		31.5	1.03 (0.88–1.21)	
Most of time or always	8.3	26.3	1.35 (1.00–1.83)		27.9	0.92 (0.73–1.15)	
**Having lunch or dinner with parents/guardian**				0.121			<0.001
≥5 days of week	60.2	18.3	1		26.9	1	
1 to 4 days of week	7.5	21.3	1.16 (0.81–1.67)		31.0	1.15 (0.91–1.45)	
<1 day of week, weekly, or never	32.3	22.9	1.25 (1.01–1.55)		35.3	1.31 (1.15–1.49)	
**Parents or guardians understand problems and concerns (past 30 days)**				0.676			0.013
Most of time or always	27.1	20.5	1		26.0	1	
Sometimes	19.6	20.6	1.00 (0.77–1.31)		29.2	1.12 (0.92–1.37)	
Never or rarely	53.4	18.8	0.92 (0.74–1.15)		32.6	1.25 (1.07–1.47)	
**Physically assaulted by an adult at family** **(past 30 days)**				0.236			0.132
Not once	57.7	18.3	1		29.4	1	
1–3 times	21.0	20.2	1.10 (0.85–1.44)		30.5	1.04 (0.89–1.21)	
≥4 times	21.4	22.0	1.20 (0.97–1.49)		34.9	1.19 (1.01–1.40)	
**Bullied or bullied by schoolmates** **(past 30 days)**				0.012			0.002
Never or rarely	39.3	16.5	1		27.2	1	
Sometimes	45.2	21.4	1.30 (1.05–1.62)		30.9	1.14 (0.99–1.30)	
Most of time or always	15.5	23.7	1.44 (1.10–1.88)		37.2	1.37 (1.15–1.63)	
**Reason/cause of colleagues intimidating or humiliating (past 30 days)**				<0.001			<0.001
My color or race	8.4	20.0	1		19.5	1	
My religion	6.0	13.4	0.67 (0.35–1.29)		29.2	1.50 (0.85–2.64)	
The appearance of my face	12.2	12.9	0.64 (0.36–1.15)		21.6	1.10 (0.66–1.84)	
The appearance of my body	15.9	43.7	2.18 (1.45–3.29)		46.4	2.37 (1.52–3.71)	
My sexual orientation	7.7	23.1	1.15 (0.72–1.86)		37.3	1.91 (1.11–3.27)	
My region	2.3	28.9	1.45 (0.78–2.67)		33.3	1.71 (0.78–3.72)	
Others	47.5	19.7	0.99 (0.65–1.49)		31.6	1.62 (1.04–2.51)	

**Table 5 children-10-01393-t005:** Association between body feelings and independent variables against Brazilian adolescents who report sexual abuse (*n* = 4088).

Variables	Total%	Negative Feeling about the Body					
		Male			Female		
		Prevalence (%)	RP Gross (IC 95%)	*p*	Prevalence (%)	RP Gross (IC 95%)	*p*
**Hungry because did not have enough food at home (past 30 days)**				0.007			0.055
Never or rarely	76.4	20.2	1		34.2	1	
Sometimes	15.4	23.2	1.15 (0.89–1.48)		35.3	1.03 (0.89–1.20)	
Most of time or always	8.3	31.6	1.56 (1.19–2.05)		42.6	1.25 (1.05–1.48)	
**Having lunch or dinner with parents/guardian**				0.039			<0.001
≥5 days of week	60.3	19.7	1		31.1	1	
1 to 4 days of week	7.5	22.6	1.15 (0.81–1.62)		38.6	1.24 (1.02–1.51)	
<1 day of week, weekly, or never	32.1	25.6	1.30 (1.06–1.59)		40.2	1.29 (1.15–1.45)	
**Parents or guardians understand problems and concerns (past 30 days)**				0.758			<0.001
Most of time or always	27.0	20.5	1		26.6	1	
Sometimes	19.7	21.1	1.03 (0.79–1.33)		30.8	1.16 (0.95–1.41)	
Never or rarely	53.3	22.2	1.08 (0.87–1.34)		40.1	1.51 (1.30–1.75)	
**Physically assaulted by an adult at family** **(past 30 days)**				0.008			0.010
Not once	57.7	19.0	1		33.0	1	
1–3 times	20.9	20.5	1.08 (0.83–1.40)		37.4	1.13 (0.99–1.29)	
≥4 times	21.4	26.0	1.37 (1.12–1.67)		40.9	1.24 (1.07–1.43)	
**Bullied or bullied by schoolmates** **(past 30 days)**				<0.001			<0.001
Never or rarely	39.3	15.7	1		29.4	1	
Sometimes	45.2	21.7	1.38 (1.11–1.73)		36.6	1.24 (1.09–1.41)	
Most of time or always	15.6	35.7	2.27 (1.81–2.87)		45.0	1.53 (1.31–1.78)	
**Reason/cause of colleagues intimidating or humiliating (past 30 days)**				0.002			<0.001
My color or race	8.5	19.0	1		34.1	1	
My religion	6.0	15.9	0.83 (0.45–1.55)		30.8	0.90 (0.57–1.44)	
The appearance of my face	12.2	26.8	1.41 (0.88–2.26)		36.7	1.08 (0.76–1.53)	
The appearance of my body	15.9	33.1	1.74 (1.12–2.71)		55.1	1.62 (1.19–2.21)	
My sexual orientation	7.7	36.9	1.94 (1.26–2.99)		45.8	1.34 (0.90–2.01)	
My region	2.3	20.5	1.08 (0.53–2.21)		27.8	0.81 (0.37–1.81)	
Others	47.5	23.7	1.25 (0.83–1.89)		34.6	1.01 (0.75–1.38)	

**Table 6 children-10-01393-t006:** Adjusted analysis of eating disorder indicators and family and school context of Brazilian students who reported sexual abuse (Vomiting and/or use of laxatives vs. Medicines).

Variables	Vomiting and/or Use of Laxatives				Medicine or Weight Loss Product			
	Male		Female		Male		Female	
	PR adj (95% CI)	*p*	PR adj (95% CI)	*p*	PR adj (95% CI)	*p*	PR adj (95% CI)	*p*
**Hungry because did not have enough food at home (past 30 days)**		0.007		0.006		0.012		0.274
Never or rarely	1		1		1		1	
Sometimes	0.90 (0.73–1.12)		1.37 (1.10–1.72)		1.03 (0.85–1.26)		1.19 (0.89–1.58)	
Most of time or always	1.37 (1.10–1.70)		1.35 (1.02–1.79)		1.39 (1.12–1.72)		1.25 (0.88–1.79)	
**Having lunch or dinner with parents/guardian**		0.128		0.008		0.659		0.003
≥5 days of week	1		1		1		1	
1 to 4 days of week	1.06 (0.83–1.36)		1.39 (1.05–1.85)		0.97 (0.75–1.26)		1.65 (1.17–2.32)	
<1 day of week, weekly, or never	0.83 (0.68–1.01)		1.30 (1.07–1.57)		0.92 (0.77–1.10)		1.37 (1.08–1.73)	
**Parents or guardians understand problems and concerns (past 30 days)**		0.091		0.033		0.011		0.003
Most of time or always	1		1		1		1	
Sometimes	0.80 (0.66–0.98)		1.24 (0.93–1.66)		0.74 (0.60–0.90)		1.44 (0.99–2.10)	
Never or rarely	0.90 (0.77–1.06)		1.36 (1.08–1.73)		0.87 (0.75–1.01)		1.69 (1.24–2.29)	
**Physically assaulted by an adult at family** **(past 30 days)**		<0.001		<0.001		<0.001		<0.001
Not once	1		1		1		1	
1–3 times	1.61 (1.30–2.00)		1.38 (1.11–1.71)		1.74 (1.41–2.15)		1.56 (1.20–2.04)	
≥4 times	2.48 (2.10–2.92)		2.11 (1.72–2.58)		2.66 (2.25–3.13)		2.47 (1.92–3.18)	
**Bullied or bullied by schoolmates** **(past 30 days)**		0.030		<0.001		0733		<0.001
Never or rarely	1		1		1		1	
Sometimes	1.06 (0.90–1.25)		1.02 (0.83–1.26)		1.00 (0.86–1.17)		0.97 (075–1.25)	
Most of time or always	1.29 (1.06–1.56)		1.75 (1.40–2.20)		1.07 (0.89–1.30)		1.88 (1.43–2.48)	
**Reason/cause of colleagues intimidating or humiliating (past 30 days)**		0.045		0.004		0.058		0.001
My color or race	1		1		1		1	
My religion	1.17 (0.84–1.64)		1.09 (0.62–1.90)		1.17 (0.83–1.66)		1.10 (0.58–2.08)	
The appearance of my face	1.07 (0.77–1.48)		0.72 (0.45–1.17)		1.03 (0.75–1.43)		0.81 (0.47–1.40)	
The appearance of my body	0.84 (0.59–1.21)		0.90 (0.58–1.41)		1.13 (0.81–1.56)		0.95 (0.58–1.57)	
My sexual orientation	0.92 (0.65–1.29)		1.05 (0.61–1.80)		0.88 (0.62–1.25)		0.89 (0.45–1.77)	
My region	1.25 (0.86–1.82)		0.71 (0.24–2.10)		1.11 (0.74–1.67)		1.13 (0.36–3.59)	
Others	0.78 (0.58–1.04)		0.59 (0.39–0.90)		0.76 (0.57–1.03)		0.49 (0.30–0.79)	

**Table 7 children-10-01393-t007:** Adjusted analysis of eating disorder indicators and family and school context of Brazilian students who reported sexual abuse (Body Dissatisfaction vs. Negative Feeling about the Body).

Variables	Body Dissatisfaction				Negative Feeling about the Body			
	Male		Female		Male		Female	
	PR adj (95% CI)	*p*	PR adj (95% CI)	*p*	PR adj (95% CI)	*p*	PR adj (95% CI)	*p*
**Hungry because did not have enough** **food at home (past 30 days)**		0.176		0.916		0.033		0.036
Never or rarely	1		1		1		1	
Sometimes	0.86 (0.61–1.23)		1.03 (0.84–1.25)		1.13 (0.84–1.52)		1.11 (0.93–1.31)	
Most of time or always	1.33 (0.93–1.90)		0.96 (0.74–1.24)		1.52 (1.10–2.08)		1.27 (1.05–1.54)	
**Having lunch or dinner with parents/guardian**		0.035		0.001		0.009		0.001
≥5 days of week	1		1		1		1	
1 to 4 days of week	1.25 (0.86–1.80)		1.02 (0.79–1.33)		1.33 (0.93–1.90)		1.15 (0.93–1.42)	
<1 day of week, weekly, or never	1.37 (1.07–1.75)		1.30 (1.13–1.51)		1.42 (1.12–1.79)		1.29 (1.13–1.46)	
**Parents or guardians understand problems and concerns (past 30 days)**		0.904		0.020		0.848		<0.001
Most of time or always	1		1		1		1	
Sometimes	0.93 (0.69–1.26)		1.13 (0.90–1.43)		0.95 (0.71–1.28)		1.33 (1.07–1.66)	
Never or rarely	0.98 (0.77–1.26)		1.28 (1.07–1.53)		1.03 (0.81–1.32)		1.56 (1.32–1.86)	
**Physically assaulted by an adult at family** **(past 30 days)**		0.648		0.168		0.135		0.050
Not once	1		1		1		1	
1–3 times	1.10 (0.82–1.46)		0.98 (0.83–1.17)		0.95 (0.71–1.29)		1.12 (0.96–1.30)	
≥4 times	1.11 (0.87–1.43)		1.19 (0.98–1.43)		1.23 (0.98–1.55)		1.21 (1.02–1.43)	
**Bullied or bullied by schoolmates** **(past 30 days)**		0.032		0.025		<0.001		<0.001
Never or rarely	1		1		1		1	
Sometimes	1.27 (0.99–1.63)		1.10 (0.94–1.29)		1.27 (0.98–1.64)		1.22 (1.06–1.41)	
Most of time or always	1.47 (1.08–2.00)		1.31 (1.08–1.59)		2.28 (1.75–2.95)		1.51 (1.27–1.79)	
**Reason/cause of colleagues intimidating or humiliating (past 30 days)**		<0.001		<0.001		0.043		<0.001
My color or race	1		1		1		1	
My religion	0.62 (0.29–1.35)		1.57 (0.84–2.93)		1.00 (0.50–1.98)		1.00 (0.59–1.70)	
The appearance of my face	0.57 (0.28–1.16)		0.86 (0.48–1.56)		1.37 (0.78–2.42)		0.98 (0.64–1.51)	
The appearance of my body	1.99 (1.23–3.23)		2.01 (1.20–3.36)		1.62 (0.96–2.74)		1.50 (1.02–2.21)	
My sexual orientation	1.08 (0.62–1.88)		1.84 (1.00–3.38)		1.86 (1.11–3.12)		1.37 (0.88–2.13)	
My region	1.62 (0.84–3.17)		1.35 (0.51–3.61)		1.29 (0.61–2.71)		0.62 (0.21–1.83)	
Others	0.93 (0.57–1.52)		1.44 (0.87–2.38)		1.10 (0.68–1.81)		0.97 (0.66–1.42)	

## Data Availability

Additional information or data can be requested by e-mail from the corresponding authors.

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
