# Peer review of "Indicators of Eating Disorders in Sexually Abused Brazilian Adolescents: Family and School Contexts"

_children, 2023, doi:10.3390/children10081393_

Round 1

Reviewer 1 Report

The abstract is insufficient (relevant information is not included).

The key words are not adequate, as well as being insufficient.

Data is very out of date.

In addition, they do not use the latest documentary edition.

This is a documentary analysis study, which does not justify the high number of authors.

Errors in citations and references.

The novelty of the study is not perceived.

English needs to be revised.

Author Response

Responses to Reviewer' Comments

Manuscript Number: 2429803

Title: Indicators of eating disorders in sexually abused Brazilian adolescents: family and school contexts

Dear Editor Carrie Cao,

We would like to express our gratitude to you, the editor, and the three reviewers for their thoughtful and in-depth comments concerning our manuscript. Your suggestions helped us improve the quality of our paper. We carefully considered every comment and made all the appropriate changes. Reviewer comments are detailed first, and our responses to the points raised follow (italics). We have highlighted the changes made to the body of the manuscript using the red color in this document and manuscript.

Editor: Dear authors, your article is interesting due to the context in which the study takes place. However, I believe that substantial revisions are necessary, before being able to resume the revision of the contribution. It is necessary to better distinguish in the sample the correlations between the different types of eating disorders and the presence of abuse in the life history. Furthermore, the conclusions need to be better argued. Otherwise everything is too generic. After these revisions you can re-submit the article. Good work!

Author´s response: Thanks for your positive feedback. We have carefully observed each comment from you and, based on them, we improved all sections of our manuscript, including Materials and Methods and Conclusions.

Reviewer #1:

The abstract is insufficient (relevant information is not included).

The key words are not adequate, as well as being insufficient.

Author´s response: Thanks for your careful review. We agree with your observation. So, we rebuilt the abstract, including the relevant information. We reviewed the key words too.

“Abstract: Eating disorders, characterized by abnormal eating behaviors, are among a wide variety of psychiatric conditions that mainly affect children and adolescents. These disorders have a multifactorial origin and can be associated with restrictive diets, negative feelings, harmful family relationships, and posttraumatic stress. Thus, this study’s objective was to evaluate the association between indicators of eating disorders and family and school contexts in Brazilian adolescents who previously experienced sexual abuse and examine the findings based on sex. Data from the National School Health Survey were utilized. Among 102,301 students between 11–19 years of age, 4,124 reported having experienced sexual abuse and were included in this study. Self-report questionnaires were used to assess participants’ health status and the presence of risk behaviors, which were examined through multivariate analysis using a Poisson regression model. The results indicated positive relationships between self-induced vomiting, laxative misuse, and other purgative methods and infrequent meals with family, hunger, and the presence of violence in students’ daily lives, regardless of sex (p < 0.05). In addition, body dissatisfaction and negative feelings about one’s body were associated with having been bullied or teased by schoolmates for both sex (p < 0.05). Distant relationships with parents were associated with purgative methods and body dissatisfaction among female students (p < 0.05). In conclusion, body dissatisfaction, negative feelings about one’s body, laxative misuse, self-induced vomiting, and purgative methods were found to be associated with factors in family and school contexts such as hunger, infrequent meals with family, family violence, distant relationships with parents, and bullying at school in adolescents who have previously experienced sexual abuse.

Keywords: Eating disorders; laxatives; adolescent’s health; sexual violence; sexual crimes; sexual assault victims; family context, school context, bullying”

Data is very out of date. In addition, they do not use the latest documentary edition.

Author´s response: We understand your concern. This population study is the largest epidemiological survey of adolescents in Brazil and is carried out each three years. However, Brazil has suffered from intense delays in carrying out most of its epidemiological surveys in recent years, due a conservative government. Despite that, past editions have not shown intense changes in the data profile, indicating similarities. This reality reinforces the need for complex and interdisciplinary public health policies and programs, which are subsidized from the analysis of these epidemiological data.   

This is a documentary analysis study, which does not justify the high number of authors.

Author´s response: This work was developed by a team of undergraduate and master's students, with the participation of advisors and professors from related disciplines. These were smaller works that were grouped for publication, with a view to not fragmenting the data. It is relevant to mention that our study is in accordance with the ICMJE (International Committee of Medical Journal Editors) and on their Recommendations for the Conduct, Reporting, Editing and Publication of Scholarly Work in Medical Journals.

Moreover, the number of authors is at the mean of epidemiological studies.

Errors in citations and references.

Author´s response: Thanks for your in-depth review. We made some mistakes in citations and reference. In this sense, we reviewed all references and made the necessary corrections.

The novelty of the study is not perceived.

Author´s response: Thanks for your careful review. We have improved the writing “about the novelty of our study, as follow:

Page 2, lines 77-89: “EDs and sexual abuse may have an intense and permanent impact on adolescent physical and mental health [24,25]. To date, few studies have aimed to verify associated factors or indicators of EDs in adolescents in family and school contexts using a representative national sample [24]. Several studies have been limited to small clinical or community samples, included only one type of EDs, or been aimed toward only the female population, making it difficult to obtain information on this topic [15]. Understanding the factors associated with indicators of EDs related to family and school contexts is essential to better understand sexual abuse survivors and provide them with effective treatment for resulting in positive long-term mental health outcomes [26]. Nevertheless, supporting health-promoting actions and improving adolescents’ quality of life according to sex differences remains important. Therefore, the present study aimed to evaluate the association between ED indicators and family and school contexts in Brazilian adolescents who have experienced sexual abuse and examine these associations based on sex.”

English needs to be revised.

Author´s response: Thanks for your review. We carry out an English native review for a full manuscript (https://www.editage.com.br/)

Reviewer #2:

Thank you for the opportunity to review this very interesting work. I hope you will find my comments useful.

Author´s response: Thanks for your review and your positive feedback. We have carefully observed each comment from you and, based on them, we improved this manuscript.

  1. The aspects of not enough food in the home may need some further context. A normative reading of that data causes one to question the role that poverty or other social determinants of health are at play here.

Author´s response: Thanks for your review. This appointment was really necessary. We included new sentences at the discussion section, as follow:

Page 12, lines 298-303: “In the present study, hunger caused by not having not sufficient food at home was associated with self-induced vomiting, laxative use and purgative methods, body dissatisfaction, and negative feelings about one’s body in both sexes. Notably, it is a marker of food and nutritional insecurity in households, characterizing itself as a social determinant of health variable [48]. Food and nutritional insecurity has been associated with disordered eating patterns in previous studies [48-50].”

  1. In the introduction, given that all of the respondents appear to have a cell phone, the literature might consider the impact of social media on body image issues which can act as reinforcers to body image issues

Author´s response: Thank you for your thoughtful comments and for your interest in our article. We appreciate your suggestion to explore the relation between the use of a cell phone (social media) and body image issues. We agree that this could be a valuable area for further research. Unfortunately, when we were designing the outline of this study, we did not take into consideration these comparisons suggested by you, which we now recognize as relevant. Therefore, we will take your suggestion into consideration for future research.

  1. at 2.3 (line 137) there is the use of diagnostic terminology - it is quite unclear to me how the diagnosis was made. DSM 5 has quite specific criteria on an individual basis and thus, the linkage between your data and diagnostic criteria is quite unclear. I suggest that behavioral clusters be considered as opposed to diagnostic labels which may be quite inaccurate.

Author´s response: We really agree with your suggestion. We made the change as suggested, as follow:

Page 03, lines 139-145: Dependent variables used in the data analysis included self-induced vomiting and/or laxative misuse, use of medication or weight loss products, body dissatisfaction, and negative feelings about one’s body, which characterize behavioral clusters common in EDs. Socioeconomic and school- and family-related variables were considered independent variables. Dependent variables were analyzed using descriptive statistics and Wald's chi-square association (bivariate analysis). Independent variables were analyzed using a Poisson regression model with robust variance (crude analysis).

  1. There is a need for some English language editing - not extensive but there are places where it is a bit awkward

Author´s response: Thanks for your review. We carry out an English native review for a full manuscript.

These are minor issues where the English just needs to be edited. Examples The abstract does not flow well - line 25  the objective OF this study;

line 27 was used to ninth - remove used and replace with given

line 103 - used to students  - s/b used with students

Author´s response: Thanks for your in-depth review. We made the changes as suggested. We also carry out an English native review for a full manuscript.

Reviewer #3:

I appreciate the opportunity to review the article entitled: Indicatives of eating disorders in sexually abused Brazilian adolescents.

Author´s response: Thanks for your feedback. We have carefully observed each comment from you and, based on them, we improved our manuscript.

About this work, I have these suggestions: 1.- The article focuses its study on different variables, such as the school and family context, but these important variables are not indicated in the title of the work, which means that the title is not entirely informative. Perhaps the authors can reconsider a broader title.

Author´s response: We completely agree with your suggestion. So, we changed the title including school and family context, as follow:

“Indicators of eating disorders in sexually abused Brazilian adolescents: family and school contexts”

 2.- The data studied are already 8 years old. Authors should explain why more recent data are not used, and why this age does not affect the validity and generality of the results.

Author´s response: We understand your concern. This population study is the largest epidemiological survey of adolescents in Brazil and is carried out each three years. However, Brazil has suffered from intense delays in carrying out most of its epidemiological surveys in recent years, due a conservative government. Despite that, past editions have not shown intense changes in the data profile, indicating similarities. This reality reinforces the need for complex and interdisciplinary public health policies and programs, which are subsidized from the analysis of these epidemiological data.     

3.- The variable eating disorders is usually used to define disorders clearly established in a diagnostic manual such as DSM 5 or ICD 11, such as Anorexia Nervosa or Bulimia Nervosa. In this article, these disorders are not evaluated or diagnosed, but an estimate of the existence of disorders is made based on some self-reported items, without validation as a diagnostic test. Perhaps the authors can consider not using the clinical term Eating Disorders and use another term that is not related to a clearly defined disorder.

Author´s response: We really agree with your consideration. For this, we used in the title the word “indicators”. Based on that, we made changes in all manuscript. Follow some examples:

Page 2, lines 54-58: “Body dissatisfaction; overeating; restrictive dieting; negative feelings such as guilt, sad-ness, fear, and anxiety; and parental influence may indicate an ED, especially if two or more are present [11-13].”

Page 2, lines 77-87: “EDs and sexual abuse may have an intense and permanent impact on adolescent physical and mental health [24,25]. To date, few studies have aimed to verify associated factors or indicators of EDs in adolescents in family and school contexts using a representative national sample [24]. Several studies have been limited to small clinical or community samples, included only one type of EDs, or been aimed toward only the female population, making it difficult to obtain information on this topic [15]. Understanding the factors associated with indicators of EDs related to family and school contexts is essential to better understand sexual abuse survivors and provide them with effective treatment for resulting in positive long-term mental health outcomes [26]. Nevertheless, supporting health-promoting actions and improving adolescents’ quality of life according to sex differences remains important.”

Page 3, lines 139-141: “Dependent variables used in the data analysis included self-induced vomiting and/or laxative misuse, use of medication or weight loss products, body dissatisfaction, and negative feelings about one’s body, which characterize behavioral clusters common in EDs.”

Page 13, lines 370-379: “A relationship exists between indicators of EDs, such as self-induced vomiting, laxative misuse and other purgative methods, body dissatisfaction, and negative feelings about one body and low perceived parental care, hunger, and violence experienced at home in both male and female adolescents. In the school context, body dissatisfaction and negative feelings toward one’s body were associated with having been bullied or teased by schoolmates. The involvement of parents and guardians with these adolescents is a protective factor against both violence and indicators of EDs, particularly in female adolescents. Therefore, adolescents who experience physical, psychological, and sexual abuse may be more dissatisfied with their bodies compared with their peers, which may con-tribute to the development of EDs.”

4.- As a complete clinical diagnosis of the subjects has not been made, a classification of Anorexia Nervosa, Bulimia Nervosa, or Binge Eating Disorder cannot be made (line 137). The authors should modify this aspect.

Author´s response: We really agree with your note as answer above. We changed the sentence, as follow:  

Page 3x, lines 139-143: “Dependent variables used in the data analysis included self-induced vomiting and/or laxative misuse, use of medication or weight loss products, body dissatisfaction, and negative feelings about one’s body, which characterize behavioral clusters common in EDs. Socioeconomic and school- and family-related variables were considered independent variables.”

5.- I recommend carrying out a more detailed analysis, in the discussion, about the processes and intermediate variables that relate the previous experience of sexual abuse and the increased risk of disorders related to eating. In the same way, indicate how preventive intervention could be made in these processes or variables. This would give the article a very interesting practical perspective.

Author´s response: Thank you very much for your feedback. As suggested, we added these appointments in the discussion, as follow:

Page 11, Lines 258-265: “The population in this study had reported previously experiencing sexual abuse, which is an intense trauma and intermediate variable for EDs [24,38-41]. A recent meta-analysis that evaluated 29,272 individuals with EDs and 1,679,385 controls noted evidence suggesting prior sexual abuse as a risk factor for BN and an association between appearance-related teasing victimization and any ED [24]. Despite the lack of knowledge regarding the relationship between these factors [24], suggest binge and dissociation behaviors, signs of EDs, are important to trauma survivors [40]. This association has implications for ED prevention and treatment [42].”

Page 12, Lines 349-360: “A comprehensive understanding of the indicators of EDs in family and school contexts among adolescents who have experienced sexual abuse may help clinicians and survivors address indicators of disordered eating and weight-related outcomes as well as identify intervention strategies for preventing injuries and ensuring necessary treatments [40]. Epidemiological data on adolescents from developing countries such as Brazil can shed light on this topic, reinforcing the need for complex and interdisciplinary public health policies and programs to guide the direction of public health [40]. For example, the link between sexual abuse and EDs has implications for ED treatment and prevention. There-fore, in treatment, EDs should not be emphasized as an individual pathology, as this reinforces the original trauma. Instead, the theme should be approached using methods that shape the ability of adolescents to respond to, make sense of, and resist trauma while living healthy lives [39].”

Reviewer 2 Report

Thank you for the opportunity to review this very interesting work. I hope you will find my comments useful.

1. The aspects of not enough food in the home may need some further context. A normative reading of that data causes one to question the role that poverty or other social determinants of health are at play here. 

2. In the introduction, given that all of the respondents appear to have a cell phone, the literature might consider the impact of social media on body image issues which can act as reinforcers to body image issues

3. at 2.3 (line 137) there is the use of diagnostic terminology - it is quite unclear to me how the diagnosis was made. DSM 5 has quite specific criteria on an individual basis and thus, the linkage between your data and diagnostic criteria is quite unclear. I suggest that behavioral clusters be considered as opposed to diagnostic labels which may be quite inaccurate.

4. There is a need for some English language editing - not extensive but there are places where it is a bit awkward

These are minor issues where the English just needs to be edited. Examples The abstract does not flow well - line 25  the objective OF this study;

line 27 was used to ninth - remove used and replace with given

line 103 - used to students  - s/b used with students

Author Response

(The authors gave the same response as above.)

Reviewer 3 Report

I appreciate the opportunity to review the article entitled: Indicatives of eating disorders in sexually abused Brazilian

adolescents. About this work, I have these suggestions:

1.- The article focuses its study on different variables, such as the school and family context, but these important variables are not indicated in the title of the work, which means that the title is not entirely informative. Perhaps the authors can reconsider a broader title.

2.- The data studied are already 8 years old. Authors should explain why more recent data are not used, and why this age does not affect the validity and generality of the results.

3.- The variable eating disorders is usually used to define disorders clearly established in a diagnostic manual such as DSM 5 or ICD 11, such as Anorexia Nervosa or Bulimia Nervosa. In this article, these disorders are not evaluated or diagnosed, but an estimate of the existence of disorders is made based on some self-reported items, without validation as a diagnostic test. Perhaps the authors can consider not using the clinical term Eating Disorders and use another term that is not related to a clearly defined disorder.

4.- As a complete clinical diagnosis of the subjects has not been made, a classification of Anorexia Nervosa, Bulimia Nervosa, or Binge Eating Disorder cannot be made (line 137). The authors should modify this aspect.

5.- I recommend carrying out a more detailed analysis, in the discussion, about the processes and intermediate variables that relate the previous experience of sexual abuse and the increased risk of disorders related to eating. In the same way, indicate how preventive intervention could be made in these processes or variables. This would give the article a very interesting practical perspective.

Author Response

(The authors gave the same response as above.)

Round 2

Reviewer 1 Report

Not all questions have been resolved, and problems remain with respect to:

- Data is very out of date.

- In addition, they do not use the latest documentary edition (the work should use the most recent data available).

- This is a documentary analysis study, which does not justify the high number of authors (more questioning of the quality of work with students).

It has improved.

Author Response

[Children] Manuscript ID: children-2429803 

Dear Editor Anastasia Yu

Assistant Editor

We would like to express our gratitude to you, the editor, to the acceptation from the Reviewer #3, as well as to the Reviewers #1 and #2 for their second-round comments. Your suggestions helped us improve the quality of our paper. We carefully considered every reviewer comments. Reviewer comments are detailed first, and our responses to the points raised follow.

Reviewer #1:

Not all questions have been resolved, and problems remain with respect to:

- Data is very out of date. In addition, they do not use the latest documentary edition (the work should use the most recent data available)

Author´s response: Thanks for your careful review. The data are considered actual and not out of date. The database Pense 2015 needs to be explore in more in-depth way for several health outcomes, as for the variables “eating disorders” and “sexual abused”. So, we are contributing with the better understanding of this scenario. Moreover, this analysis is essential for future comparations between other documentary editions (previous and future editions). If this does not occur, a lack of analysis will be given and will prevent comparative analysis.

This reality reinforces the need for complex and interdisciplinary public health policies and programs, which are subsidized from the analysis of these epidemiological data.

It is important to note that this population study is the largest epidemiological survey of adolescents in Brazil and is carried out each three years. However, Brazil has suffered from intense delays in carrying out most of its epidemiological surveys in recent years, due a conservative government.

- This is a documentary analysis study, which does not justify the high number of authors (more questioning of the quality of work with students).

Author´s response: Thanks for your comment. All authors made considerable work on this manuscript. All authors participated effectively on the intellectual construction of this study.

We attend all the statements from the ICMJE (International Committee of Medical Journal Editors) and on their Recommendations for the Conduct, Reporting, Editing and Publication of Scholarly Work in Medical Journals.

Moreover, the number of authors is at the mean of epidemiological studies.

Reviewer 2 Report

Thank you for the attention to detail in the revisions. Clearly a significant effort went into this.

One final thought - instead of using the word "sex" it is now more  common to refer to gender. In addition, there is more thought about the terms "male identifying" and "female identifying" - these terms reflect the growing recognition of gender fluidity which may be prevalent in an ED population where body dysmorphia and gender identity can be at play. This might be an area for minor revision

Author Response

[Children] Manuscript ID: children-2429803 

Dear Editor Anastasia Yu

Assistant Editor

We would like to express our gratitude to you, the editor, to the acceptation from the Reviewer #3, as well as to the Reviewers #1 and #2 for their second-round comments. Your suggestions helped us improve the quality of our paper. We carefully considered every reviewer comments. Reviewer comments are detailed first, and our responses to the points raised follow.

Reviewer #2:

Thank you for the attention to detail in the revisions. Clearly a significant effort went into this.

One final thought - instead of using the word "sex" it is now more  common to refer to gender. In addition, there is more thought about the terms "male identifying" and "female identifying" - these terms reflect the growing recognition of gender fluidity which may be prevalent in an ED population where body dysmorphia and gender identity can be at play. This might be an area for minor revision.

Author´s response: Thanks for your careful review. About the use of the term “sex” instead “gender”, the reason is that in our study we considered the participants in a biologic way and we just evaluate “boys” and “girls”. If we would like to evaluate “gender”, more that “male” and “female” options should be given to the participants answering. For the present study and aims, our choice is in accordance with the literature (Diamond, 2022; Krieger, 2003).

Diamond M. Sex and Gender are Different: Sexual Identity and Gender Identity are Different. Clinical Child Psychology and Psychiatry. 2002;7(3):320-334. doi:10.1177/1359104502007003002

Krieger N. Genders, sexes, and health: what are the connections—and why does it matter? International Journal of Epidemiology. 2003; 652-657, doi:10.1093/ije/dyg156
